# Ending the HIV epidemic using National HIV Behavioral Surveillance (NHBS): Recommendations based on DC model

Suparna Das[1]*, Richard Medina[2], Emily Nicolosi[2], Anya Agopian[3], Irene Kuo[3], Jenevieve Opoku[1], Adam Allston[1], Michael Kharfen[1]

1 Strategic Information Division, HIV/AIDS, Hepatitis, STD, and TB Administration (HAHSTA), District of Columbia Department of Health, Government of the District of Columbia, Washington, DC, United States of America, 2 Department of Geography, University of Utah, Salt Lake City, Utah, United States of America, 3 Milken Institute of Public Health, Department of Epidemiology, George Washington University, Washington, DC, United States of America

☯ These authors contributed equally to this work.

* Suparna.das@dc.gov

## Abstract

### Introduction

Social network strategies have been used by health departments to identify undiagnosed cases of HIV. Heterosexual cycle (HET4) of National HIV Behavioral Surveillance (NHBS) is a social network strategy implemented in jurisdictions. The main objectives of this research are to 1) evaluate the utility of the NHBS HET cycle data for network analysis; 2) to apply statistical analysis in support of previous HIV research, as well as to develop new research results focused on demographic variables and prevention/intervention with respect to heterosexual HIV risk; and 3) to employ NHBS data to inform policy with respect to the EHE plan.

### Method

We used data from the 2016 NHBS HET4 (DC). A total of 747 surveys were collected. We used the free social-network analysis package, GEPHI, for all network visualization using adjacency matrix representation. We additionally conducted logistic regression analysis to examine the association of selected variables with HIV status in three models representing 1) demographic and economic effects, 2) behavioral effects, and 3) prevention-intervention effects.

### Results

The results showed 3% were tested positive. Seed 1 initiated the largest networks with 426 nodes (15 positives); seed 4 with 273 nodes (6 positives). Seed 3 had 35 nodes (2 positives). All 23 HIV diagnoses were recruited from 4 zip-codes across DC. The risk of testing positive was higher among people high-school dropouts (Relative Risk (RR) (25.645); 95 CI % 5.699, 115.987), unemployed ((4.267); 1.295, 14.064), returning citizens ((14.319); 4.593, 44.645). We also found in the final model higher association of pre-exposure

**Data Availability Statement:** Data cannot be shared publicly because of PHI and PII DC and federal/GW Institutional Data Access / Ethics Committee. There are legal and ethical restrictions

to the data being made available publicly. The data can be made available upon request (contact via manyadm@gwu.edu or acastel@gwu.edu).

**Funding:** The study was supported by the National Center for HIV/AIDS, Viral Hepatitis, STD, and TB Prevention. The funders had no role in study design, data collection and analysis, decision to publish, or preparation of the manuscript.

**Competing interests:** The authors have declared that no competing interests exist.

prophylaxis (PrEP) awareness among those tested negative ((4.783); 1.042, 21.944) and HIV intervention in the past 12 months with those tested positive ((17.887); 2.350,136.135).

## Conclusion

The network visualization was used to address the primary aim of the analysis-evaluate the success of the implementation of the NHBS as a social network strategy to find new diagnoses. NHBS remains one of the strongest behavioral supplements for DC's HIV planning activities. As part of the evaluation process our analysis helps to understand the impact of demographic, behavioral, and prevention efforts on peoples' HIV status. We strongly recommend other jurisdictions use network visualizations to evaluate the efficacy in reaching hidden populations.

## Introduction

Identifying persons with undiagnosed Human Immunodeficiency Virus (HIV) infection and linking them to medical care and prevention services is a national priority [1, 2]. Fittingly, some present-day HIV research focuses on behavioral risks and new diagnoses using social network analysis (SNA) for knowledge extraction, as HIV transmits primarily through social interaction, either with direct personal contact (e.g., sexual activity), or through an intermediate activity (e.g., intravenous drug use) [3–5]. SNA has proven to be a valuable approach for HIV research with respect to HIV risk and drug use [6], transmission [7], influences on homeless youth [8], and venue based risk [9], among many others. This study first applies SNA and statistical regression models to identify HIV risk factors and evaluate the effectiveness of the National HIV Behavioral Surveillance (NHBS) strategy in heterosexual HIV diagnoses, and second provides recommendations to improve the implementation of the NHBS program across the United States to guide the Ending the HIV Epidemic–A plan for America (EHE).

   We select the Washington, D.C. metro area for this study as it aligns well with national goals and trends. With an HIV prevalence of 1.8%, Washington, D.C. was selected as one of the target hotspots in the EHE. The NHBS program was developed by the Centers for Disease Control and Prevention (CDC), a component of the National HIV/AIDS Strategy for the United States in 2002, to help state and local health departments establish and maintain a surveillance system to monitor selected behaviors and prevention services among groups at the highest risk for HIV infection [10]. The NHBS uses social network data to assist in developing prevention efforts to reduce the spread of HIV [11]. It has played a major role in providing information on HIV related behavioral risks and HIV testing to U.S. jurisdictions. NHBS outcomes improve the understanding of HIV risk and testing behaviors, which are used to implement, as well as evaluate programs for communities [8].

   The importance of this research is two-fold. First, there have been numerous studies using the *men who have sex with men* (MSM) and *people who inject drugs* (IDU) cycle data from NHBS surveys; however, research conducted on the *heterosexual* (HET) cycle has been limited to behavioral analyses alone, such as the anal intercourse risks associated with reporting [12], unprotected sex with casual/exchange partners [13], and substance abuse [13]. Heterosexual contact is the second most common route of HIV transmission, estimated to have accounted for approximately 24% of the new infections diagnosed in 2018 [14]. We add to the literature on heterosexual (HET cycle) HIV risk by including models of demographic and

prevention/intervention variables. Previous analyses have revealed the association of individual-level HIV-associated risk factors with increased risk of HIV [15], yet they fail to address the gaps in prevention-intervention, which is critical in planning the EHE. Second, the implications of NHBS as a significant policy tool for the EHE plan has been missing, which we offer here.

This research is the first effort to map NHBS HET data networks and their characteristics, to identify any evidence of HIV positive network clustering, and the first to use HET NHBS data to make recommendations for HIV policy implementation to achieve goals set out in the EHE. The main objectives of this research are to 1) evaluate the utility of the NHBS HET cycle data for network analysis, which is used by the CDC to identify new HIV positives; 2) to apply statistical analysis in support of previous HIV research, as well as to develop new research results focused on demographic variables and prevention/intervention with respect to heterosexual HIV risk; and 3) to employ NHBS data to inform policy with respect to the EHE plan. Our methods include social network visualization and regression analysis.

## Data and methods

### NHBS survey data

The National HIV Behavioral Surveillance (NHBS) program administered by the D.C. Department of Health and George Washington University under CDC directives, includes separate HIV based surveys and HIV testing among high-risk populations, which include men who have sex with men (MSM cycle), people who inject drugs (IDU Cycle), and heterosexuals with increased risk of HIV (HET Cycle).

We use 2016 NHBS HET Cycle data, specifically heterosexual cycle 4 (HET4), which was the fourth round of heterosexual data collection. In general, HET data were collected within 22 metropolitan statistical areas (MSAs) across the United States. The MSAs were selected based on their relatively high number of people living with HIV/AIDS. Survey participants were recruited by respondent-driven sampling (RDS). RDS is initiated with a limited number of "seed" participants who are purposefully chosen through formative research. These individuals were then given 3–5 coupons to recruit social connections into the study. Recruitment continued until the sample size was met, or the end of the data collection period was reached. A total of 747 surveys were collected through the NHBS HET cycle.

Eligibility for the heterosexual cycle of NHBS was restricted to men and women between 18 and 60 years old, who had not previously participated in 2016, were residents of a study MSA, were able to complete the survey in English or Spanish, were able to provide informed consent, and reported having vaginal or anal sex in the past 12 months with an opposite sex partner. Potential participants who reported ever injecting drugs or male-to-male sex in the past 12 months were not eligible to participate. For each survey cycle, an anonymous standardized questionnaire was used to collect information about HIV-associated behaviors, specifically sexual behaviors, substance use, HIV testing, and use of HIV prevention services [16].

For NHBS HET cycles, formative assessments include activities that identify and characterize High Risk Areas (HRAs). HRAs are geographic areas within the MSA where heterosexuals are at higher risk for HIV infection compared to other geographic areas within the MSA. HRAs are defined as areas with high rates of poverty. HRAs are used to identify appropriate locations for storefronts or van locations during survey implementation and to identify seeds. Locating the HRAs requires project sites to obtain geographic data, identify areas of high poverty, and map the HRAs using a Geographic Information System (GIS). Officials within the state health department are involved in the process of identifying HRAs, because it entails handling confidential data. HRA identification is described in greater detail in the NHBS Formative Assessment Manual [10]. Because the focus is on high-risk areas the data do not serve as a

random sample spatially or demographically. A large portion of the data are collected from a relatively smaller sample of locations from a large sample of Black/African Americans. While this can be an issue in interpreting the results of this research, we admit that our results are best applied within high-risk areas and high-risk populations.

Seeds must be residents of High-Risk Areas (HRAs). Since many social ties are formed among individuals who live on the same street or in the same neighborhood, it is important that seeds be residents of HRAs, so that recruitment begins in areas most likely to have a high proportion of low socio-economic status (SES) residents. HRA residency is assessed during the survey. Potential seeds who do not live in an HRA may participate in NHBS HET but will not be eligible to recruit others.

The NHBS funded by CDC and implemented by DC DOH with George Washington University. NHBS IRB for DC DOH was 2014 3 and GWU it is 12331.

## Digital network representation

To represent the connections collected between people in the NHBS HET survey, we used personal social network representations based on index individuals, or "seeds". The SURID field contains ID numbers for each node in the dataset, including the seeds, and c1-c5 fields that identify social connections between each of the individuals in the dataset. Records are duplicated, such that survey responding nodes found in the c1-c5 fields are all represented in the SURID field. Null values in the dataset representing non-responses are marked -1. Using the SURID (node IDs) and c1-c5 (adjacent node connections and their IDs) fields, network visualizations were constructed for three seed nodes. Given seven initial seeds, only six returned connections, and only three returned meaningful network representations. The social network analysis package, GEPHI, was used for all network visualizations. The NHBS data contain many other fields representing demographic, behavioral, and health data, as described in the Center for Disease Control's, NHBS HET4 CAPI Reference Questionnaire (CRQ) document [17].

The final compiled network dataset included selected data fields that were appended to the network node data. This allowed for the inclusion of node attributes, such as HIV Positive identifiers ('EVERPOS', 'finrslt'), as shown in the network figures. The 'EVERPOS" variable in the data set represents whether or not the participant has ever tested positive for HIV. We tallied finrslt and EVERPOS variables to get the final HIV positive number for the analysis.

For networks initiated by Seeds 1, 3, and 4, a Louvain modularity algorithm for community detection was applied, such that the networks are segmented into "modules" [18]. This algorithm maximizes the within module connectivity and minimizes the between module connectivity.

## Multinomial logistic regression analysis

A multinomial logistic regression was applied to model outcome dependent variables and examine the associations between demographic, behavioral, and prevention-intervention variables and HIV. Multinomial logistic regression analysis is used when the dependent variable is comprised of more than two categories. We calculated a relative risk ratio (RR) for the associations to better interpret the results. We did not use imputation or other substitution methods were not used. In logistic regression, a logistic transformation of the odds (referred to as the logit) serves as the depending variable.

Where:

p = the probability that a case is in a particular category,

exp = the exponential,

a = the constant of the equation and,

b = the coefficient of the predictor or independent variables

The likelihood ratio test is based on -2LL ratio. Significance at the 0.05 level or lower means the model with the predictors is significantly different from the one with the constant only (all 'b' coefficients being zero). It measures the improvement in fit that the explanatory variables make compared to the null model [19].

The variables were selected, and models were constructed in coordination with D.C.'s Ending the HIV Epidemic plan (EHE)

**Outcome variable.** The outcome variable for the regression model is HIV status, and is coded into the following three categories: 0 "did not test" 1 "tested negative" and 2 "tested positive."

**Predictor variables.** The predictor variables were coded into categories for the regression analysis. All predictor variables were obtained from the NHBS data set. The primary aim of the regression model was to evaluate variables that are being considered or implemented through various programs by DC for Ending the HIV Epidemic by 2030, so the variables for the regression analysis were selected based on priorities of policies that are implemented or in the process of being implemented. The predictor variables were divided into three categories: 1. demographic and economic variables, 2. behavioral variables, and 3. prevention-intervention variables for HIV. The variables were selected to evaluate the impact of policies that are currently implemented or in the process of being implemented in DC. The models measure the changes in the degree of association with the addition of each model's variable set at each of the three model iterations. The first model evaluates the association of HIV outcomes with demographic and economic variables. The second model (behavioral) adds variables including crack cocaine use, mental health conditions, jail term, sexual debut age and casual partners. These are priorities of DC's Ending the HIV Epidemic Plan for 2030. The final model (prevention-intervention) uses variables including technology, pre-exposure prophylaxis awareness and HIV interventions.

Multicollinearity tests were conducted on the models using variance inflation factors (VIFs) and tolerance. If a VIF value exceeded 4.0 or had a tolerance of less than 0.2, the variable was not included in the model as multicollinearity would be indicated [18]. Coefficients with 95% confidence intervals (CIs) were presented in the tables and the relationships were considered statistically significant at *p-value* $< 0.05$.

## Analysis and results

### Descriptive statistics

Table 1 shows the characteristics of the participants in the study. Out of 747 participants for the NHBS HET 4 cycle, approximately 73% (n = 545) tested negative, 24% (n = 178) had unknown status, which likely means they opted out of the test, and 3% (n = 24) were found HIV positive (Table 1). Out of those positives approximately, 42% (n = 10) were above ages 50 and 55% (n = 12) had education of grade 9 through 11. We combined two variables EVERPOS (Survey question: "Have you ever tested positive for HIV, that is, do you have HIV?") and finrslt (Survey question: 'Final HIV result of those who were tested') to identify the HIV positives in the study. EVERPOS showed 15 people who knew they were HIV positive, and the total number of people identified as HIV positives were 24. The number of potential new diagnoses through the cycle were 8. The testing and reporting processes were anonymous, so we cannot confirm if they were new positives (Table 1).

Table 2 shows the characteristics of selected variables for seeds and connections. Approximately 71.43% (n = 5) of the seeds recruited in the sample were male and 28.57% (n = 2) were

**Table 1. Descriptive characteristics of NHBS HET 4 cycle DC by HIV status.**

| Participants characteristics | Not Tested | Tested Negative | Tested Positive | Total |
|---|---|---|---|---|
| **Education** | | | | |
| No information | 139(78.09) | 0(0.00) | 0(0.00) | 139(18.61) |
| Grades 1 through 8 | 14(7.87) | 138(25.32) | 5(25.32) | 157(21.02) |
| Grades 9 through 11 | 18(10.11) | 301(55.23) | 12(55.23) | 331(44.31) |
| Grade 12 and higher | 7(3.93) | 106(19.45) | 7(19.45) | 120(16.06) |
| Total | 178(100) | 545(100) | 24(100) | 747(100) |
| **Age** | | | | |
| Age less than 20 (Ref) | 23(12.92) | 41(7.52) | 1(4.17) | 65(8.70) |
| Ages between 26 to 30 | 33(18.54) | 166(30.46) | 2(8.33) | 201(26.91) |
| Ages between 31 and 40 | 14(7.87) | 100(18.35) | 4(16.67) | 118(15.80) |
| Ages 41 and 50 | 27(15.17) | 108(19.82) | 7(29.17) | 142(19.01) |
| Age more than 50 | 81(45.51) | 130(23.85) | 10(41.67) | 221(29.59) |
| Total | 178(100) | 545(100) | 24(100) | 747(100) |
| **Gender** | | | | |
| Male | 105(58.99) | 288(52.84) | 12(50.00) | 405(54.22) |
| Female | 71(39.89) | 257(47.16) | 12(50.00) | 339(45.38) |
| Transgender | 2(1.12) | 0(0.00) | 0(0) | 2(0.27) |
| Total | 178(100) | 545(100) | 24(100) | 747(100) |
| **Race** | | | | |
| American Indian | 2(1.12) | 1(0.18) | 0(0.0) | 3(0.40) |
| Asian | 0(0.00) | 1(0.18) | 0(0.0) | 1(0.13) |
| Black/African American | 168(94.38) | 526(96.51) | 24(100.00) | 719(96.25) |
| Native Hawaiian | 0(0.00) | 1(0.18) | 0(0.0) | 1(0.13) |
| White | 8(4.49) | 14(2.57) | 0(0.0) | 22(2.95) |
| **Total** | 178(100) | 543(100) | 24(100) | 747(100) |

female. Roughly 57.14 (n = 4) were between ages 21 and 30 and 85.71% (n = 6) were Black/African American.

## Descriptive mapping

Fig 1 maps the distribution of 747 study participants and 23 HIV positives in the study based on residence zip codes. Geocoding was conducted using Maptitude, a proprietary GIS software developed by Caliper Corporation. All 23 HIV positives were in 4 zip codes–zip code number 20002 had one HIV case, 20030 had 11 HIV cases, 20019 had 6 HIV cases, and 20032 had 5 HIV cases. Based on the maps we found a higher number of participants were recruited from Wards 7 and 8 (Fig 1).

## Network visualizations

The network visualizations show each network segmented by community (module). Seeds 1, 3, and 4 had the most extensive connected networks, and also larger numbers of participants who tested positive. Seed 1, who was male, initiated a network through which 15 positives were identified, with one branch including 6 positives. A larger branch extending from Seed 1 recruited 3 clusters of positives. There was one transgender person identified in this network (Fig 2).

Seed 4, who was also male, initiated the second largest network (Fig 2). This network included 6 positives total and recruited one transgender person. We did not identify multiple

Table 2. Descriptive characteristics of HET cycle 4 of seeds and associates.

| Gender | Associate | Seed | Total |
|---|---|---|---|
| Male | 400(54.13) | 5(71.43) | 405(54.22) |
| Female | 338(45.74) | 2(28.57) | 340(45.52) |
| Transgender | 2(0.27) | 0(0) | 2(0.27) |
| Total | 739(100) | 7(100) | 747(100) |
| **Age groups** | | | |
| Age less than 20 | 63(8.51) | 2(28.57) | 65(8.70) |
| Age between 21 and 30 | 197(26.62) | 4(57.14) | 201(26.91) |
| Age between 31 and 40 | 117(15.81) | 1(14.29) | 118(15.80) |
| Age between 41 and 50 | 142(19.19) | 0(0) | 142(19.01) |
| Age 50 and above | 221(29.86) | 0(0) | 221(29.59) |
| Total | 740(100) | 7(100) | 747(100) |
| **Race** | | | |
| American Indian | 3(0.41) | 0(0) | 3(0.40) |
| Asian | 1(0.14) | 0(0) | 1(0.13) |
| Black/African America | 684(92.81) | 6(85.71) | 690(92.37) |
| Native Hawaiian | 9(1.22) | 1(14.29) | 10(1.34) |
| Refuse to answer | 40(5.43) | 0(0) | 40(5.35) |
| Unknown | 0(0) | 6(42.86) | 6(0.40) |
| Total | 737(100) | 7(100) | 747(100) |

network clusters of HIV positives for Seed 4. Seed 3 (Fig 2) was initiated by a male positive and had two other positives identified in the personal network. The size of the network (35 nodes) is considerably smaller compared to networks for Seeds 1 and 4. Seeds 2, 5, and 7 were substantially smaller. Seed 2 was the largest of the three, recruiting 9 individuals, primarily females. None of the three smaller networks included any HIV positive nodes.

## Characterizing the sample using multinomial logistic regression

We constructed three iterative models to characterize the data and observe changes as variables were added. Model 1 showed ages 31–40 (RR: 6.902; 95% CI: 2.580, 18.460) were at higher risk of testing negative followed by ages 26–30 (RR: 4.445; 95% CI: 1.884, 10.486). Education with 9 and 11 grades (RR: 18.11; 95% CI: 7.525, 43.588) were also found to be at higher risk of HIV followed by those with education between 1st and 8th grade (RR: 17.511; 95% CI: 9.906, 30.952) for all who tested negative. Unemployment showed association (RR: 10.607; 95% CI: 5.233, 21.498) with those who tested negative. The likelihood ratio (LR) chi-square of Model 1, -352.29 with a *p-value* of 0.0001, suggests that our model was statistically significant, compared to the null model with no predictors (Table 3). The AIC for Model 1 is 744.587 (Table 3).

In Model 2 we added the behavioral variables to evaluate association with the risk of HIV. The association of risk of HIV with education and age decreased with the addition of the behavioral risks for both who tested positive and negative. Females (RR: 2.445; 95% CI: 1.444, 4.148) were found to be at higher odds of HIV for negatives. With the addition of the behavioral risks in Model 2 we found increased risk of testing positive with lower education–Grades 1 through 9 (RR: 15.913; 95% CI: 4.629, 54.699) and grades 9th through 11 (RR: 25.645; 95% CI: 5.699, 115.987). We also found increased risk of testing positive (RR: 4.267; 95% CI: 1.295, 14.064) with unemployment with the addition of behavioral variables. Increased risk of HIV for people who used crack cocaine (RR: 6.799; 95% CI: 1.714, 26.978) compared to those who tested negative and used cocaine (RR: 5.564; 95% CI: 2.040, 15.179). Our model also showed

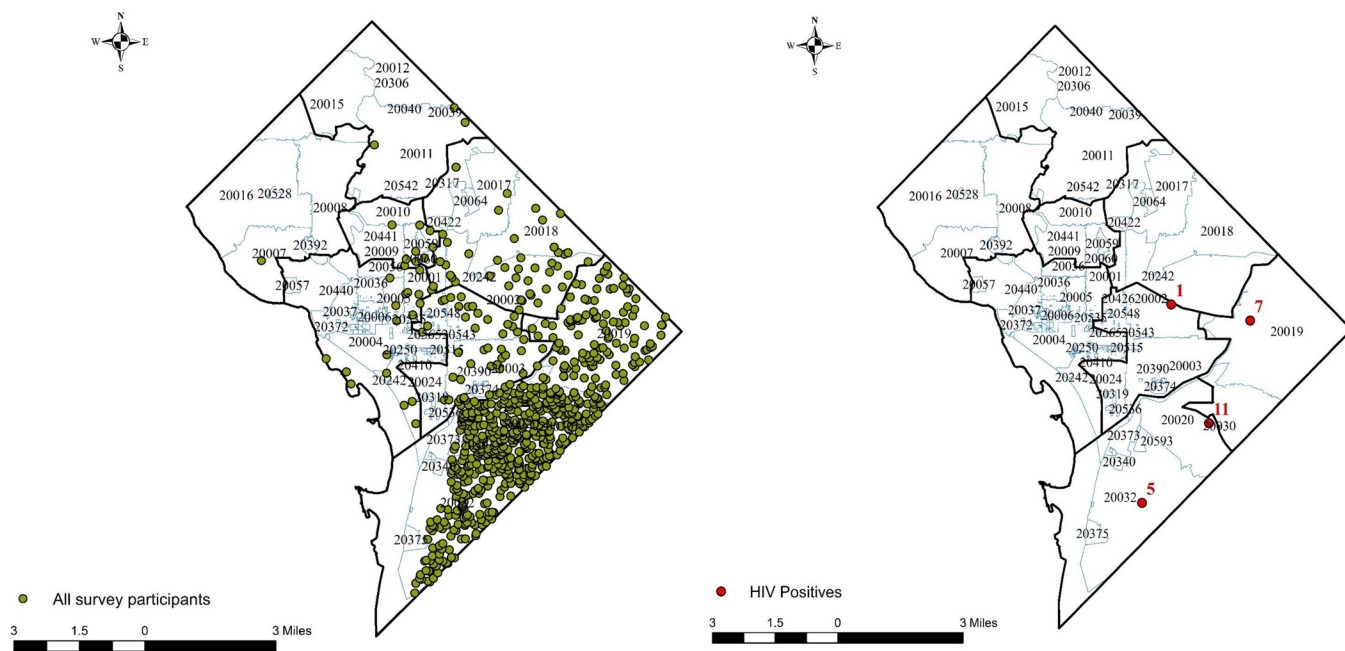

**Fig 1. Distribution of HET4 cycle participants and HIV positives in DC.** The green dots show the distribution of the survey recruitment. The numbers in red for HIV positives designate the total number of cases in zip code.

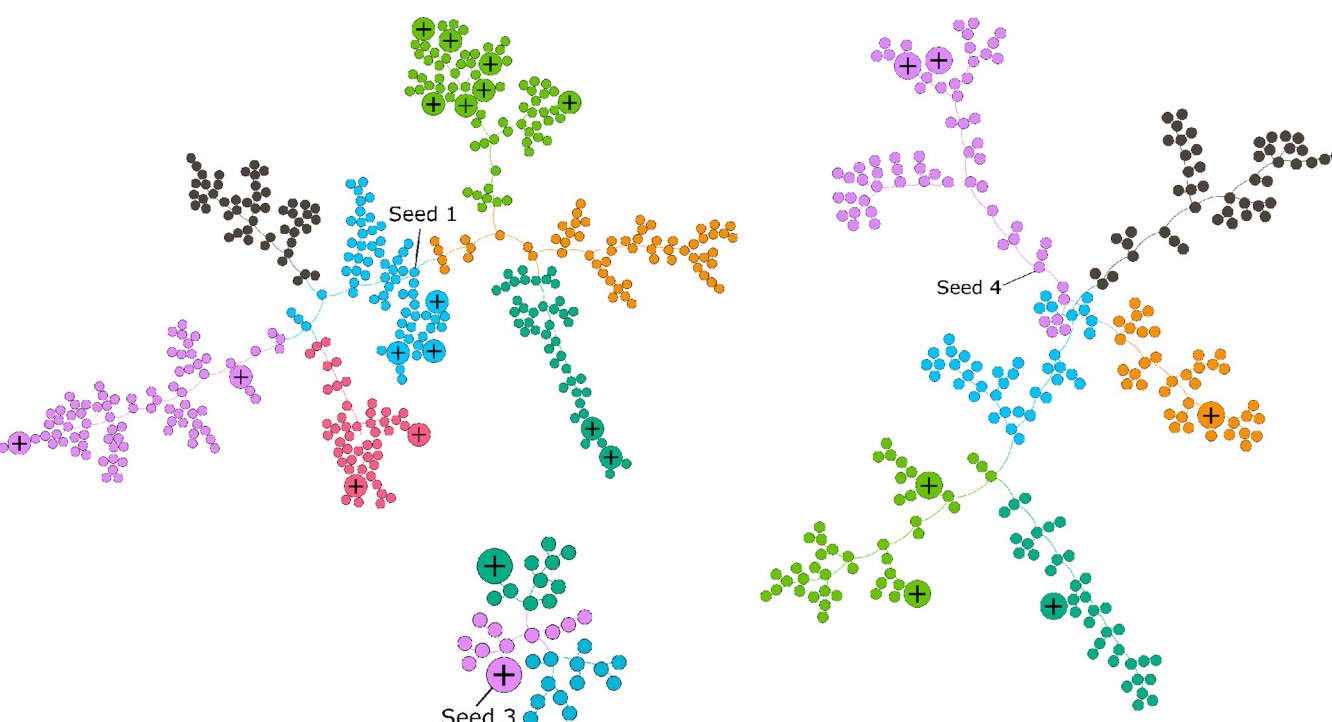

**Fig 2. Seeds 1, 3, and 4, the largest networks of HET4 cycle.** The positive signs indicate the HIV positives in the networks.

**Table 3. Regression analysis of HIV status outcome.**

| Base outcome for all models (Not Tested) | Model 1 Tested Negative RR | STD ERROR | 95% CI | Model 1 Tested Positive RR | STD ERROR | 95% CI | Model 2 Tested Negative RR | STD ERROR | 95% CI | Model 2 Tested Positive RR | STD ERROR | 95% CI | Model 3 Tested Negative RR | STD ERROR | 95% CI | Model 3 Tested Positive RR | STD ERROR | 95% CI |
|---|---|---|---|---|---|---|---|---|---|---|---|---|---|---|---|---|---|---|
| **Demographic Variables** | | | | | | | | | | | | | | | | | | |
| Male (Ref) | | | | | | | | | | | | | | | | | | |
| Female | 1.390 | 0.317 | 0.897–2.181 | 1.586 | 0.735 | ###–3.933 | 2.45** | 0.66 | 1.44–4.15 | 3.02 | 1.61 | 1.06–8.60 | 2.37** | 0.658 | 1.380–4.080 | 3.379* | 1.9 | 1.12–10.18 |
| **Age Groups** | | | | | | | | | | | | | | | | | | |
| Ages between 18 and 25 (ref) | | | | | | | | | | | | | | | | | | |
| Ages between 26 to 30 | 4.45*** | 1.95 | 1.88–10.49 | 1.89 | 2.46 | 0.15–24.06 | 3.14* | 1.49 | 1.23–7.98 | 1.08 | 1.43 | 0.08–14.34 | 3.38* | 1.69 | 1.27–8.99 | 0.93 | 1.29 | 0.06–13.91 |
| Ages between 31 and 40 | 6.90*** | 3.46 | 2.58–18.46 | 9.74 | 11.93 | 0.88–107.36 | 4.87** | 2.64 | 1.69–14.09 | 4.75 | 5.96 | 0.41–55.58 | 5.20** | 2.91 | 1.74–15.55 | 4.87 | 6.33 | 0.38–62.35 |
| Ages 41 and 50 | 4.10* | 1.85 | 1.69–9.91 | 9.77 | 11.30 | 1.01–94.32 | 2.51 | 1.24 | 0.95–6.61 | 4.06 | 4.85 | 0.39–42.20 | 2.97* | 1.53 | 1.08–8.17 | 3.50 | 4.35 | 0.31–39.85 |
| Age more than 50 | 2.46* | 1.03 | 1.08–5.58 | 6.66 | 7.54 | 0.72–61.35 | 1.21 | 0.58 | 0.48–3.08 | 2.28 | 2.74 | 0.22–23.96 | 1.59 | 0.79 | 0.60–4.22 | 1.54 | 1.94 | 0.13–18.22 |
| **Education** | | | | | | | | | | | | | | | | | | |
| Grades 1 through 8 | 17.51*** | 5.09 | 9.91–30.95 | 23.43 | 14.15 | 7.17–76.50 | 10.94*** | 3.52 | 5.82–20.53 | 15.91*** | 10.02 | 4.63–54.70 | 9.76*** | 3.18 | 5.15–18.50 | 19.85*** | 13.36 | 5.31–74.26 |
| Grades 9 through 11 | 18.11*** | 8.12 | 7.53–43.59 | 38.25 | 27.88 | 9.17–159.62 | 9.67*** | 4.68 | 3.74–24.99 | 25.65*** | 19.75 | 5.67–#### | 7.49*** | 3.67 | 2.87–19.55 | 34.44*** | 28.06 | 6.98–170.02 |
| Grades 12 or GED | 11.42* | 12.20 | 1.41–92.69 | 0.00 | 0.08 | 0.00–0.00 | 3.35 | 3.77 | 0.37–30.43 | 0.004 | 0.02 | 0.00–0.00 | 2.19 | 2.48 | 0.24–20.07 | 0.00 | 0.02 | 0.00–0.00 |
| Some Collge (Ref) | | | | | | | | | | | | | | | | | | |
| Unemployed | 10.61*** | 3.82 | 5.23–21.50 | 8.16 | 4.73 | 2.62–25.43 | 5.12*** | 2.02 | 2.37–11.08 | 4.27* | 2.60 | 1.30–14.06 | 5.27*** | 2.12 | 2.39–11.60 | 3.50 | 2.32 | 0.96–12.80 |
| **Behavioral Variables** | | | | | | | | | | | | | | | | | | |
| Use Crack Cocaine | | | | | | | 5.56*** | 2.85 | 2.04–15.18 | 6.80** | 4.78 | 1.71–26.98 | 4.66** | 2.41 | 1.69–12.86 | 7.02** | 5.13 | 1.68–29.37 |
| Diagnosed Mental health Condition | | | | | | | 2.58** | 0.99 | 1.21–5.49 | 0.47 | 0.35 | 0.11–2.06 | 2.04 | 0.88 | 0.87–4.76 | 0.41 | 0.32 | 0.08–1.95 |
| Been to jail (returning citizens in DC EHE) | | | | | | | 7.13*** | 2.61 | 3.48–14.60 | 14.32*** | 8.31 | 4.59–44.64 | 5.09*** | 1.99 | 2.36–10.97 | 15.67*** | 10.45 | 4.24–57.92 |
| Sexual debut as a minor | | | | | | | 0.14** | 0.12 | 0.03–0.70 | 0.18 | 0.20 | 0.02–1.68 | 0.147* | 0.12 | 0.03–0.71 | 0.18 | 0.21 | 0.02–1.79 |
| 1 partner in the last 12 months (ref) | | | | | | | | | | | | | | | | | | |
| 2 - 3 partners in the last 12 months | | | | | | | 6.99** | 4.84 | 1.80–27.13 | 7.472* | 7.19 | 1.13–49.23 | 7.61** | 5.27 | 1.96–29.57 | 8.477* | 8.32 | 1.24–58.05 |
| More than 4 partners in the last 12 months | | | | | | | 1.18 | 0.43 | 0.58–2.39 | 1.79 | 1.10 | 0.54–5.95 | 1.22 | 0.45 | 0.60–2.52 | 1.94 | 1.25 | 0.55–6.84 |
| **Intervention and Prevention Variables** | | | | | | | | | | | | | | | | | | |
| Use Technology for health information (Computer, Tablet, Smartphone) | | | | | | | | | | | | | 1.91 | 0.79 | 0.85–4.29 | 0.46 | 0.32 | 0.12–1.79 |
| Heard of PrEP | | | | | | | | | | | | | 4.783* | 3.72 | 1.04–21.94 | 0.91 | 1.23 | 0.07–12.80 |
| HIV intervention in the past 12 months | | | | | | | | | | | | | 2.60 | 2.33 | 0.44–15.14 | 17.887** | 18.52 | 2.35–136.14 |
| AIC | 744.587 | | | | | | 641.377 | | | | | | -273.80 | | | | | |
| BIC | 836.908 | | | | | | 789.091 | | | | | | 623.61 | | | | | |
| Log Likelihood | (-)352.294*** | | | | | | (-)288.6885*** | | | | | | 799.017*** | | | | | |

increased risk among positive tested returning citizens (RR: 14.319; 95% CI: 4.593, 44.645) compared to those who tested negative (RR: 7.132; 95% CI: 3.485, 14.598) (Table 3).

In Model 3 we added prevention-intervention variables, which are currently implemented, or being considered for implementation in DC. While risk of demographic association with testing positive increased for high school drop-out groups (grades 1$^{st}$– 8$^{th}$ and 9$^{th}$ - 11$^{th}$). This may indicate the efficacy of prevention-intervention strategies. Pre-exposure prophylaxis awareness showed higher association to those who were at risk but tested negative (RR: 4.783; 95% CI: 1.042, 21.944), while HIV intervention showed association with those who tested positive (RR: 17.887; 95% CI: 2.350, 136.135). The AIC for Model 3 is -273.803 (Table 3).

## Discussion

The network visualization here addresses some of the utility of the NHBS as a survey mechanism, which has the capacity to recruit and test people who are at high risk of HIV using a social network strategy. Generalized testing strategies have become less effective in HIV diagnoses and the NHBS was implemented in jurisdictions to reach the "hidden" population (i.e., those for whom no sampling frame exists or whose members engage in stigmatized or illegal activities, making them reticent to divulge information that may compromise their privacy) [6]. There have been several analyses based on the NHBS data, unfortunately none of them address the success of the network implementation to find HIV diagnoses, which lies at the core of the implementation. Based on our network analysis we strongly recommend other jurisdictions employ network visualizations to evaluate the efficacy in reaching their own hidden populations. However, the structure of the social network data may offer little more than visualization. Still, visualizations often prove to be powerful tools. We recommend implementing CDC's Social Network strategy to recruit hidden population through community-based organizations. The network visualization shows evidence of some recruitment clustering for HIV positives. We recommend re-evaluating the NHBS recruitment strategy to expand to areas that include more diverse populations in D.C.

This analysis identifies valuable results that can play an important role in HIV prevention-intervention planning and future data collection efforts. The HIV recruitment sample maps identify 24 positives residing in four zip codes, which suggests spatial clusters of HIV and supports previous studies that have identified HIV clusters in D.C. [19, 20]. That said, these recruits were primarily from High-Risk Areas (HRA) identified based on poverty and HIV diagnoses maps provided by the Health Department and were found based on spatially biased data. Three zip codes 20019, 20020, and 20032 had the highest number of recruits in the networks (see Table 1).

As part of the evaluation process our analysis helps to understand the impact of demographic, behavioral, and prevention efforts on peoples' HIV status. It is reported that Black women aren't always aware of their higher risk for HIV which stems from lack of public health awareness [20], we recommend higher recruitment and testing of Black women including strategies for PrEP awareness. Black communities traditionally have a high degree of social mixing between higher and lower risk individuals, which means that they are more likely to have a partner with a history of higher risk [20, 21]. Black women are relatively more susceptible to poverty, unstable housing, and unemployment, potentially increasing the likelihood of participation in sex trade, and making them more vulnerable to HIV [22]. We acknowledge that these situations are likely the result of structural racism. Black women also have a higher likelihood of experiencing violence or trauma. This is an important finding for D.C.'s EHE plan where increased engagement of Black women with HIV services and PrEP. We strongly recommend that the plan prioritizes Black women and their access to health care.

Among this HET sample we found that association of education did not change the HIV outcomes with behavior or interventions. The sample analysis also showed that crack cocaine, jail time, and a larger number of partners increased the risk of HIV for HETs in D.C. These behavioral issues may be significant as DC shapes the plan to end the HIV epidemic (EHE). As a part of the opioid use disorder and harm reduction initiative, has been planning and implementing several programs which include the Opioid Learning Institute led by the HealthHIV, opioid awareness campaign and education, and as well as Medication Assisted Treatment (MAT) and Substance Use Disorder Treatment (SUD). It is well established that substance use, including crack cocaine, can create a cycle in which people quickly exhaust their resources and turn to other ways to acquire the drug, including trading sex for drugs or money. This increases HIV risk, yet we lack programs that help mitigate it. We recommend collaboration with the Department of Behavioral Health (DBH) to develop behavioral treatment guidelines for people at risk or with HIV. The guidelines will assist providers with helping patients initiate abstinence and avoid relapse to cocaine use. These guidelines should include contingency management, cognitive behavioral therapy, and motivational interviewing. We also recommend integrating behavioral health screening for people who may be at risk of HIV and being prescribed PrEP.

Our results support that incarceration increases the risk of HIV. Unfortunately, this well-documented outcome often ignores the racial aspect. In the United States, Black and Latino/Hispanic men of lower socio-economic status are disproportionately incarcerated [23]. We recommend programs that may assist incarcerated individuals with prevention and treatment adherence in collaboration with the Metropolitan Police Department Detention Center. D.C. plans to incorporate programs to evaluate and monitor HIV treatment adherence among incarcerated individuals through the returning citizens program.

NHBS is planning on implementing home-based testing post COVID-19. Jurisdictions across the U.S. are considering home-based HIV testing, which will be shipped to those at-risk following completion of an online form. These jurisdictions must be aware of the digital divide that may have a significant impact on HIV outcomes, particularly during the current pandemic. We recommend that for prevention and intervention, we continue to engage with traditional community-based organizations and Disease Intervention Specialists in these areas of lower internet access.

## NHBS limitations and recommendations for further sampling and research

First, there is a possibility of sampling bias. This is clear in the spatial distribution of those surveyed. A chain-referral sampling method will be overly sampled within social networks, as defined by the seeds, while larger regional patterns of HIV infection will remain unknown.

Second, the identified positives were held anonymous; thus, they could not be confirmed as new HIV diagnoses. Owing to the CDC recommended pre-condition of selecting criteria of areas, the generalizability becomes difficult to assess. Our results also suggest that the HIV diagnoses may have been previous positives. This is not recommended for optimized social network strategy which has the capacity to identify new positives. We strongly recommend that CDC consider evaluating recruitment guidelines for NHBS.

We worry that current NHBS implementation strategies may not have enough considerations for Hispanic population, making them vulnerable to HIV. We recommend recruitment expansion to areas with higher Latinx population.

Social network analysis in this case is limited, given that traditional calculations would prove to be of little resource. Each node is connected to 1–6 other nodes, where direction through the network, also, does not provide much information. Network data collected in this

way will not have much clustering utility aside from visual, as their hierarchical nature restricts complex community identification. However, there remains an important use, to identify individual social network branches in which higher rates of infections are occurring. Characterizing those branches may prove valuable given that some spatial information exists within the database. Still, there are questions about sampling bias with respect to the non-random choices made to pass on coupons to the next person and its impact on finding the undiagnosed HIV.

Despite these limitations NHBS remains one of the strongest behavioral supplements for D. C.'s HIV planning activities. We recommended planning for a more widely sampled social network study to study sexual-sociospatial networks in D.C. as it can provide more information than solely social or spatial analyses alone. The study and results would be valuable in planning activities for Ending the HIV Epidemic—A plan for America.

## Supporting information

**S1 Table. Characteristics of networks.**
(DOCX)

## Author Contributions

**Conceptualization:** Suparna Das.

**Data curation:** Suparna Das, Richard Medina, Anya Agopian, Jenevieve Opoku.

**Formal analysis:** Suparna Das, Richard Medina, Emily Nicolosi.

**Funding acquisition:** Irene Kuo, Jenevieve Opoku, Michael Kharfen.

**Investigation:** Suparna Das.

**Methodology:** Suparna Das.

**Project administration:** Suparna Das, Anya Agopian.

**Resources:** Suparna Das, Irene Kuo, Jenevieve Opoku, Adam Allston, Michael Kharfen.

**Software:** Suparna Das, Richard Medina, Emily Nicolosi.

**Supervision:** Suparna Das.

**Validation:** Suparna Das.

**Visualization:** Suparna Das, Richard Medina, Emily Nicolosi.

**Writing – original draft:** Suparna Das, Richard Medina.

**Writing – review & editing:** Suparna Das, Richard Medina, Irene Kuo.

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
