## [Decision Letter · Decision Letter 0]

8 Apr 2021

PONE-D-21-00506

Ending the HIV Epidemic using National HIV Behavioral Surveillance (NHBS): Recommendations based on DC model.

PLOS ONE

Dear Dr. Das,

Thank you for submitting your manuscript to PLOS ONE. After careful consideration, we feel that it has merit but does not fully meet PLOS ONE’s publication criteria as it currently stands. Therefore, we invite you to submit a revised version of the manuscript that addresses the points raised during the review process.

Your manuscript was reviewed by one expert in the field.  Unfortunately, many potential reviewers could not accept the invitation.  Nevertheless, comments generated by the reviewer are comprehensive enough and identify many important  issues in your submission. Please carefully consider the attached comments and provide your thorough responses.

We look forward to receiving your revised manuscript.

Kind regards,

Yury E Khudyakov, PhD

Academic Editor

PLOS ONE

Journal Requirements:

"NO The funders had no role in study design, data collection and analysis, decision to

publish, or preparation of the manuscript."

5. Please include your tables as part of your main manuscript and remove the individual files. Please note that supplementary tables (should remain/ be uploaded) as separate "supporting information" files

Reviewers' comments:

Reviewer's Responses to Questions

**Comments to the Author**

1. Is the manuscript technically sound, and do the data support the conclusions?

Reviewer #1: Partly

2. Has the statistical analysis been performed appropriately and rigorously? 

Reviewer #1: No

3. Have the authors made all data underlying the findings in their manuscript fully available?

Reviewer #1: No

4. Is the manuscript presented in an intelligible fashion and written in standard English?

Reviewer #1: Yes

5. Review Comments to the Author

Reviewer #1: This paper used data from 2016 NHBS HET4 in DC area. Although of interest, the paper needs to be edited before consideration for publication.

The authors should be consistent in defining the objectives of the study. There is inconsistency in the abstract, i.e., primary aim was more general is making recommendations in using this NHBS database, while the conclusion stated that the aim was to evaluate the use of NHBS as a social network to find new HIV cases. The Introduction stated a third definition of primary aim.

The Introduction is difficult to follow and should be edited. The acronyms should be spelled out the first time they appear in the text. From the Introduction, it is difficult to apprehend what is known and what this particular approach will bring to the existing literature or in which way that could help drawing some recommendations for HIV prevention.

The Methods are unclear. The authors reported a mapping at census track while the results are presented at zip codes level. The logistic regression is also unclear as the hypothesis tested is not clearly defined. The regression examined the association with HIV status defined according to 3 categories. However, the results seemed to report only on tested association with HIV-positive status. The predictors were coded into binary however in the Tables, some outcomes are obviously in several categories (one excluding the other ones, e.g., age, gender, education level). It is unclear how the variables entering in the models have been identified. Typically, variables that have been showed associated with HIV status in univariate analyses should be entered in the model. The authors controlled for collinearity but considering the small sample of HIV-positive and the number of variables, the associations by chance should be controlled and tests should be applied (likelihood-ratio test, Benjamini-Hochberg correction to control for false discovery rate).

Odds ratio with CI that large questioned the level of precision of the OR and suggested sparse data. This is really difficult to interpret.

The representativity of the sample is also questionable. Almost all the sample was Black/AA. Whereas 70% of people living with HIV are black in DC area, the representativity of other race (White, Asian, Native) and ethnicity (Hispanic (Latin(x)) and non-Hispanic) is essential in order to draw any recommendation for ending the HIV epidemic. The Figure 1 also showed a lack of representativity of the DC area. There might have a rationale for it, however, the authors need to provide comments regarding limits in sample representativity.

It is unclear which interpretation of findings related to intervention could be draw and how that is informative for policy. Most of the intervention and prevention variables were probably highly correlated with other factors age, gender, education that more likely also be associated with race.

Table 3 needs to be edited. HIV status is defined according to three categories and only one OR is reported.

6. PLOS authors have the option to publish the peer review history of their article (what does this mean?). If published, this will include your full peer review and any attached files.

Reviewer #1: No

---

## [Author Response · Author response to Decision Letter 0]

17 May 2021

To

Dr. Yury E Khudyakov, PhD

Plos One

Dear Dr. Khudyakov, 

 We are pleased to resubmit our manuscript entitled “Ending the HIV Epidemic using National HIV Behavioral Surveillance (NHBS): Recommendations based on DC model” for consideration for publication in Plos One. 

 We would like to take this opportunity to thank you and the reviewer(s) for their review which substantially enhanced the quality of our manuscript. The study went through extensive IRB from DC DOH (DC DOH IRB: 2014-3) and GW IRB (121331). 

 We did not receive any funding to conduct the analysis. We do not have any conflict of interest. Please address all correspondence concerning this manuscript to me. Please let me know if you have any questions. It is always a pleasure to have you as our manuscript editor. 

Thank you

With regard,

Sincerely

Suparna Das

HIV/AIDS, STD, TB Administration

District of Columbia Department of Health 

Reviewers' comments and responses

Comments to the Author

1. Is the manuscript technically sound, and do the data support the conclusions?

Reviewer #1: Partly

2. Has the statistical analysis been performed appropriately and rigorously? 

Reviewer #1: No

3. Have the authors made all data underlying the findings in their manuscript fully available?

Reviewer #1: No. 

Response: The data may be made available by submitting data request to Dr. Adam Allston. The data has PII and PHI which needs IRB approval to be made available for any requester. 

4. Is the manuscript presented in an intelligible fashion and written in standard English?

Reviewer #1: Yes

5. Review Comments to the Author

Reviewer #1:

This paper used data from 2016 NHBS HET4 in DC area. Although of interest, the paper needs to be edited before consideration for publication.

Response:

Thank you. We have conducted major edits based on your comments.

Comment:

The authors should be consistent in defining the objectives of the study. There is inconsistency in the abstract, i.e., primary aim was more general is making recommendations in using this NHBS database, while the conclusion stated that the aim was to evaluate the use of NHBS as a social network to find new HIV cases. The Introduction stated a third definition of primary aim.

Response:

Thank you. We now discuss our objectives in detail at the end of the Introduction section and have made sure they align with the abstract.

Comment:

The Introduction is difficult to follow and should be edited. The acronyms should be spelled out the first time they appear in the text. From the Introduction, it is difficult to apprehend what is known and what this particular approach will bring to the existing literature or in which way that could help drawing some recommendations for HIV prevention.

Response:

All acronyms are now spelled out the first time they appear in the text. A brief section on existing literature has been added to the introduction, which helps the reader understand what has, or hasn’t, been done in the past. We now explain that a good amount of research has been conducted on HIV risk outside of heterosexual communities, and that research on heterosexual HIV risk has been limited to selected behavioral aspects. We modeled the changes in demographic and behavioral aspects with prevention intervention. This has never been addressed and remains imperative for Ending the HIV Epidemic (EHE) policy implementation. 

Comment:

The Methods are unclear. The authors reported a mapping at census track while the results are presented at zip codes level.

Response:

We apologize for the confusion. The text has been edited to reflect our mapping at the Zip Code level. 

Comment:

The logistic regression is also unclear as the hypothesis tested is not clearly defined. The regression examined the association with HIV status defined according to 3 categories. However, the results seemed to report only on tested association with HIV-positive status.

Response:

Thank you. We have added a much clearer discussion of our hypotheses for the statistical model into the main paper from the supplement. We updated the model to a multinomial logistic model, as we believe this fits the data and three outcomes best. This is shown in detail in Table 3. All the categories are explained in the text. 

Comment:

The predictors were coded into binary however in the Tables, some outcomes are obviously in several categories (one excluding the other ones, e.g., age, gender, education level). 

Response:

We have addressed the issue. The reference category mentioned in table 3 with additional clarifications. 

Comment:

It is unclear how the variables entering in the models have been identified. Typically, variables that have been showed associated with HIV status in univariate analyses should be entered in the model.

Response:

The variables selected for the models are based on previous research and variables suggested in DC’s Ending the Epidemic Plan. We now state this clearly in the methods section. Secondly, we are not sure what univariate analyses would add to this research. Our models were constructed based on EHE plan suggestions and the significance of our models is sufficient.

Comment:

The authors controlled for collinearity but considering the small sample of HIV-positive and the number of variables, the associations by chance should be controlled and tests should be applied (likelihood-ratio test, Benjamini-Hochberg correction to control for false discovery rate).

Response:

The sample size (non-probability sample) is 747. Hidden population do not provide sampling frames. In such cases non-probability sampling is used which is the case for NHBS. The new positives are NOT a small sample but an outcome. DC has significantly reduced new diagnoses over the years with the aim to further attain the 90/90/90/50 goals, (Dr. Das’s calculations https://www.dcendshiv.org/our-plan/numbers-were-tracking). As you can see the outcome is not small, it is what we see overall in DC. The models include likelihood ratio tests. 

Comment:

Odds ratio with CI that large questioned the level of precision of the OR and suggested sparse data. This is really difficult to interpret.

Response:

Yes, we have changed the model to a multinomial logistic regression. This seems to fit much better and makes more sense in interpretation. 

Comment:

The representativity of the sample is also questionable. Almost all the sample was Black/AA. Whereas 70% of people living with HIV are black in DC area, the representativity of other race (White, Asian, Native) and ethnicity (Hispanic (Latin(x)) and non-Hispanic) is essential in order to draw any recommendation for ending the HIV epidemic.

Response:

The burden of HIV in DC falls on the Black population. We have cited our annual report multiple times in the text. While DC is gentrifying, and hence, becoming more diverse, still the DC base population remains Black. The Black population is at the highest risk for heterosexual HIV. The data reflect populations at the highest risk for HIV. We believe that our study remains a reasonable representation of inner-city populations. The method was implemented through NHBS recruitment guidelines from Centers for Disease Control and Prevention. There may be areas where Latinx population may be recruited and we have addressed in the limitation section in the manuscript. 

Comment:

The Figure 1 also showed a lack of representativity of the DC area. There might have a rationale for it, however, the authors need to provide comments regarding limits in sample representativity.

Response:

NHBS HET cycle is recruited using social network and is a non-probability sample. The method is RDS and used to identify hidden population where sample calculation is not possible. The figure is simply the distribution of the samples. We have analyzed these data and in our study recommendations, we pose some solutions, with respect to both geography and sampling methods. There are improvements that can be made to compile more valuable datasets in the future, which we state. This is one of our main motivations for this study, and a very important one when considering the allotment of resources for future efforts. That said, our models and model results are most relevant when applied to high-risk areas and high-risk populations for EHE planning. 

Comment:

It is unclear which interpretation of findings related to intervention could be draw and how that is informative for policy.

Response:

We have added paragraphs of text in our discussion that ties each finding with a policy recommendation that is either implemented or will be implemented in the future. 

Comment:

Most of the intervention and prevention variables were probably highly correlated with other factors age, gender, education that more likely also be associated with race.

Response:

We are not sure we understand the review. Is this a statement? In that case we would need additional clarification on this assumption. The VIF tests that we ran suggest that multicollinearity in the model is not an issue which we mentioned in our manuscript. Correlation is not causation, which infers that what we are seeing in any statistical model is a relationship between the variables, not necessarily that one is causing the other, though that may be the case. 

Comment:

Table 3 needs to be edited. HIV status is defined according to three categories and only one OR is reported.

Response:

We have revised Table 3 based on your comment.

---

## [Decision Letter · Decision Letter 1]

9 Jun 2021

Ending the HIV Epidemic using National HIV Behavioral Surveillance (NHBS): Recommendations based on DC model.

PONE-D-21-00506R1

Dear Dr. Das,

We’re pleased to inform you that your manuscript has been judged scientifically suitable for publication and will be formally accepted for publication once it meets all outstanding technical requirements.

Kind regards,

Yury E Khudyakov, PhD

Academic Editor

PLOS ONE

Additional Editor Comments (optional):

Reviewers' comments:

Reviewer's Responses to Questions

**Comments to the Author**

1. If the authors have adequately addressed your comments raised in a previous round of review and you feel that this manuscript is now acceptable for publication, you may indicate that here to bypass the “Comments to the Author” section, enter your conflict of interest statement in the “Confidential to Editor” section, and submit your "Accept" recommendation.

Reviewer #1: All comments have been addressed

2. Is the manuscript technically sound, and do the data support the conclusions?

Reviewer #1: Yes

3. Has the statistical analysis been performed appropriately and rigorously? 

Reviewer #1: Yes

4. Have the authors made all data underlying the findings in their manuscript fully available?

Reviewer #1: Yes

5. Is the manuscript presented in an intelligible fashion and written in standard English?

Reviewer #1: Yes

6. Review Comments to the Author

Reviewer #1: This authors have addressed the comments of the reviewers and have edited the manuscript accordingly.

As a minor comments, the authors could delete the section where they described the model and H0/H1 hypothesis in the Methods section. This is a standard model and do not need such details.

7. PLOS authors have the option to publish the peer review history of their article (what does this mean?). If published, this will include your full peer review and any attached files.

Reviewer #1: No

---

## [Editor Report · Acceptance letter]

13 Jul 2021

PONE-D-21-00506R1 

Ending the HIV Epidemic using National HIV Behavioral Surveillance (NHBS): Recommendations based on DC model. 

Dear Dr. Das:

I'm pleased to inform you that your manuscript has been deemed suitable for publication in PLOS ONE. Congratulations! Your manuscript is now with our production department. 

Kind regards, 

on behalf of

Dr. Yury E Khudyakov 

Academic Editor

PLOS ONE